# Inspecting and Editing Knowledge Representations in Language Models

**Evan Hernandez, Belinda Z. Li & Jacob Andreas**
Computer Science & Artificial Intelligence Laboratory
Massachusetts Institute of Technology
{dez,bzl,jda}@mit.edu

## Abstract

Neural language models (LMs) represent facts about the world described by text. Sometimes these facts derive from training data (in most LMs, a representation of the word *banana* encodes the fact that bananas are fruits). Sometimes facts derive from input text itself (a representation of the sentence *I poured out the bottle* encodes the fact that the bottle became empty). We describe REMEDI, a method for learning to map statements in natural language to fact encodings in an LM's internal representation system. REMEDI encodings can be used as *knowledge editors*: when added to LM hidden representations, they modify downstream generation to be consistent with new facts. REMEDI encodings may also be used as *probes*: when compared to LM representations, they reveal which properties LMs already attribute to mentioned entities, in some cases making it possible to predict when LMs will generate outputs that conflict with background knowledge or input text. REMEDI thus links work on probing, prompting, and LM editing, and offers steps toward general tools for fine-grained inspection and control of knowledge in LMs.

## 1 Introduction

Neural language models (LMs) build implicit, structured models of the state of the world: their representations encode general knowledge (Petroni et al., 2019) and situations described in input text (Li et al., 2021). Sometimes these representations contain mistakes, resulting in errors in generated text (Fig. 1). As LMs improve, versions of these problems are likely to persist: large LM training sets contain erroneous and contradictory information, go out of date, and harbor unexpected biases (Bender et al., 2021). Even in domains where LM generation is more reliable, understanding how model-internal representations relate to output is crucial for attribution and controlled generation (Akyürek et al., 2022; Dai et al., 2022). There is thus a fundamental need for techniques that can inspect and edit LMs' knowledge, whether derived from training data or input text.

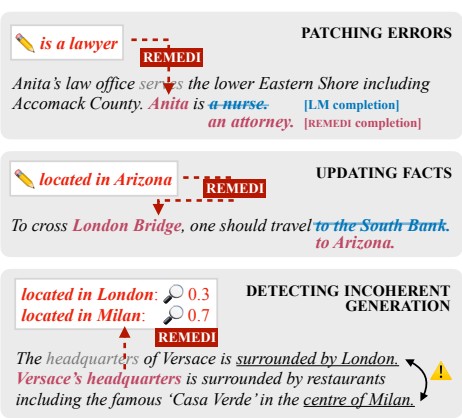

Figure 1: REMEDI can patch errors made by an LM and insert new facts with or without context provided in the prompt. It can also help detect errors before generation.

This paper introduces REMEDI (REpresentation MEDIation), a technique for discovering directions in LM-internal representation spaces corresponding to encodings of factual attributes (like *is a lawyer* in Fig. 1). When these encodings are added to LMs' representations of entities (like *Anita*), they *edit* the facts that LMs attribute to those entities—in some cases producing output that cannot be produced with a corresponding textual prompt. Encodings produced by REMEDI can also be used to *interpret* LM representations, making it possible

to probe LMs' factual knowledge, and to predict when they will generate incorrect or incoherent output.

Even when trained only to modify LMs' background knowledge, REMEDI generalizes to tasks that require querying and modifying knowledge specified in-context. Our findings thus suggest that LMs represent and integrate information from these sources in a unified manner. REMEDI offers steps towards tools that can monitor and control language generation by directly specifying facts and situations in an LM's native encoding scheme.[1]

## 2 REMEDI

**Motivations: control and interpretability**   Consider the examples from Fig. 1 (top). In the first example, the LM is **prompted** with the text *Anita's law office serves the lower Eastern Shore...*, which provides some **context** about the **entity** Anita. However, when the LM generates a continuation of this prompt, it asserts that Anita is a nurse. We term this incoherence a failure of **context integration**: information provided in the textual context has failed to alter the LM's predictions. It would be useful to identify and fix such errors, changing a model's encoding of entities like *Anita* to ensure that she is correctly described as an *attorney*. In addition to ensuring discourse coherence, it is often desirable to modify prior associations in LMs. In Fig. 1 (middle) the LM strongly associates *London Bridge* with the city of *London* because the most famous London Bridge is located there. However, there could be (and are[2]) other London Bridges, and we might wish to control an LM to make the lesser-known bridge more salient.

It is sometimes possible to achieve these goals by carefully prompting models with the right input text. But due to the non-systematic, opaque nature of prompt engineering (Jiang et al., 2020b), significant manual effort is often required to find a prompt (if one exists at all) that yields correct behavior and generalizes to new use cases.[3] Techniques for localizing generation failures within LMs' internal representations would make it possible to detect them in advance, and guide research aimed at mechanistic understanding of the relationship between LMs' internal representations and their textual outputs.

**Overview**   At a high level, our proposed approach learns how to intervene in an LM's representation space to modify the LM's knowledge about a mentioned entity (like *Anita* in Fig. 2). This intervention ultimately updates the LM's representation of the entity to encode an **attribute** (e.g., *is a lawyer*) so that the LM will generate text about the entity consistent with the new attribute. This update operation can be specified by a single vector, and is applied to the hidden representation of a single token at a single layer. Edits produced by REMEDI can also be applied out-of-context (enabling controlled generation without textual prompts). By comparing edit vectors to unedited representations, REMEDI additionally makes it possible to inspect representations of entities and attributes produced during ordinary model operation.

**Editing representations**   Assume we have a language model $p_{\text{LM}}(x)$ that assigns probabilities to strings $x$ consisting of tokens $x_{1:n}$. In this paper, $p_{\text{LM}}$ will always be an autoregressive transformer (Vaswani et al., 2017) pretrained on English text, as in the GPT family of models (Radford et al., 2019; Brown et al., 2020). These models decompose $p(x)$ into a product of next-token distributions given preceding context: $p_{\text{LM}}(x) = \prod_i p_{\text{LM}}(x_i \mid x_{1:i-1})$. Our goal is to modify an LM's internal state to cause it to generate desired text about a target entity.

Where and how should we perform this modification? LMs encode factual information in hidden representations of entity mentions: entities' states (Li et al., 2021), perceptual

---

[1]Code and data are available at `https://github.com/evandez/REMEDI`.
[2]Such as the one in Lake Havasu City, Arizona.
[3]These issues are not solved with scale: "prompt injection attacks" that cause LMs to ignore initial instructions (Perez & Ribeiro, 2022; Greshake et al., 2023) may also be viewed as failures of context integration, and might (beyond the scope of this paper) also be mitigated with better tools for directly manipulating representations of tasks rather than facts.

features (Abdou et al., 2021), and other semantic properties (Grand et al., 2018) have been shown to be *linearly decodable* from entity representations. To ensure that an LM encodes the fact *Anita is a lawyer*, it should thus suffice to find an appropriate transformation of the representation of the token *Anita*.

Formally, we denote the transformer's hidden representation for token $x_i$ in layer $\ell$ as $h_i^{(\ell)}$, and we write $p_{\text{LM}}(x \mid h_i^{(\ell)} = z)$ to mean the output of $p_{\text{LM}}$ with $h_i^{(\ell)}$ "hard-coded" to equal $z$ during the forward pass.[4] Given representations of the entity $h_{\text{entity}}$ and the target attribute $h_{\text{attr}}$, REMEDI specifically learns an **affine transformation** $F$ that returns a new entity representation $z$ according to:

$$z = F(h_{\text{entity}}, h_{\text{attr}}) = h_{\text{entity}} + W h_{\text{attr}} + b \,. \tag{1}$$

such that when $z$ replaces the entity inside the LM, the LM will generate text consistent with the target.

How should we pick $h_{\text{attr}}$, $W$ and $b$? Building on the success of linear probing approaches (Conneau et al., 2018; Belinkov & Glass, 2019), it is tempting to begin by training a *classifier* for the presence or absence of attributes. For example, following Li et al. (2021), we could take $h_{\text{attr}}$ to be the LM's own representation of an attribute (like *plays the oboe*; Fig. 2), then optimize $W$ and $b$ to predict whether an entity representation encodes the attribute:

$$p(\text{attribute} \mid \text{entity}) = \sigma(h_{\text{entity}}^{\top} W h_{\text{attr}} + b) \,. \tag{2}$$

However, even when an LM encodes information in its representations, this information may not *causally influence* subsequent generation (Ravfogel et al., 2020; Elazar et al., 2021; Ravichander et al., 2021). An effective editor must identify fact encodings that are causally linked to output.

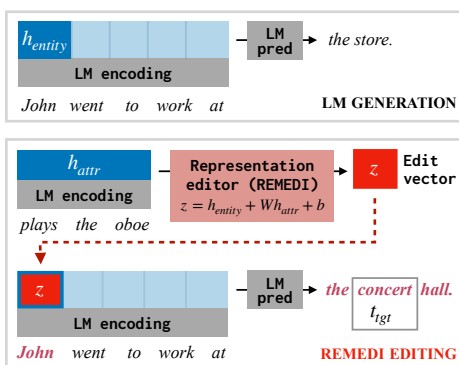

Figure 2: Illustration of REMEDI. Given an prompt (*John went to work at*) and a desired attribute (*plays the oboe*), REMEDI constructs an edit to the hidden representation of *John* that increases the probability of an appropriate completion (*the concert hall*).

**Learning Effective Edits**     REMEDI optimizes $W$ and $b$ to directly intervene in an LM. We assume access to a dataset of tuples $(x_{1:n-1}, i_{\text{entity}}, t_{\text{attr}}, t_{\text{tgt}})$, where $x_{1:n-1}$ is a textual **context** (e.g. *John went to work at*), $i_{\text{entity}}$ is the index of an entity within the context, $t_{\text{attr}}$ is the attribute to be inserted (*plays the oboe*), and $t_{\text{tgt}}$ is a **generation target**: a completion that should be assigned high probability if the attribute is applied to $x_{\text{entity}}$ (*the concert hall*). Following Li et al. (2021), we obtain a representation $h_{\text{attr}}$ by averaging the LM's encoding of $t_{\text{attr}}$. We train the editor $F$ to maximize the probability that $p_{\text{LM}}$ assigns to $t_{\text{tgt}}$ after modifying the hidden representation of $x_{\text{entity}}$ (Fig. 2):

$$\mathcal{L}_{\text{tgt}}(z) = -\log p_{\text{LM}}(x_n = t_{\text{tgt}} \mid x_{1:n-1}, h_{\text{entity}} = z) \,. \tag{3}$$

**Learning Non-Destructive Edits**     When LMs encode strong prior associations between entities and properties (e.g., in the *London Bridge* example; see Hase et al., 2021), it is necessary to remove these facts while inserting new ones. We obtain a target string $t_{\text{prior}}$ assigned a high pre-edit probability, and train $F$ to minimize the probability of $t_{\text{prior}}$:

$$\mathcal{L}_{\text{prior}}(z) = \log p_{\text{LM}}(x_n = t_{\text{prior}} \mid x_{1:n-1}, h_{\text{entity}} = z) \,. \tag{4}$$

Finally, to prevent the degenerate solution in which the language model always (and only) predicts $t_{\text{tgt}}$, we penalize the language model for changing its distributions on all tokens between the entity mention and the time at which it predicts $t_{\text{tgt}}$:

$$\mathcal{L}_{\text{KL}}(z) = \sum_{x_i}^{x_i \neq x_{\text{entity}}} D_{\text{KL}}\Big(p_{\text{LM}}(\cdot \mid x_{<i}, h_{\text{entity}} = z) \,\Big\|\, p_{\text{LM}}(\cdot \mid x_{<i})\Big) \,. \tag{5}$$

---

[4]Henceforth we will omit the layer index, but the reader should always assume all operations occur at a single layer.

Unlike the distribution over tokens at the end of the prompt, which should change dramatically under the intervention, the distribution over these intermediate tokens should not change significantly. $\mathcal{L}_{\text{KL}}$ penalizes such changes. Thus, the complete objective function is:

$$\mathcal{L}(z) = \mathcal{L}_{\text{tgt}}(z) + \lambda_1 \mathcal{L}_{\text{prior}}(z) + \lambda_2 \mathcal{L}_{\text{KL}}(z) \,, \tag{6}$$

where $\lambda_1$ and $\lambda_2$ are hyper-parameters. We evaluate REMEDI by studying its ability to control model output (Section 4) and to interpret and predict model behavior (Section 5).

## 3 Related Work

**Probing factual knowledge**   Large language models (LLMs) trained on massive text datasets have been shown to encode context-agnostic factual knowledge, which can be queried through a text prompt (Petroni et al., 2019). Most work on extracting background factual knowledge from LMs focuses on designing textual *queries* for different sources of knowledge (Richardson & Sabharwal, 2020; Peng et al., 2022). Additionally, *knowledge probes* may sometimes recover factual information even in cases when LMs do not generate truthful outputs with high probability (Burns et al., 2022).

**Probing representations of individual situations**   Neural LMs have also been shown to build representations of context-dependent knowledge. (Li et al., 2021) show that they track aspects of entity state over a discourse, and this state can be extracted from LM representations of contextualized entity tokens. Furthermore, many LMs have been (indirectly) evaluated on their ability to track context-dependent knowledge by having their performance measured on downstream *reading comprehension* tasks in wich the LM is expected to answer questions about facts within a discourse. Reading comprehension datasets such as CoQA (Reddy et al., 2019), RACE (Lai et al., 2017), and SQuAD (Rajpurkar et al., 2016) are now part of the standard evaluation suite for new LMs; and most modern LMs perform well (Brown et al., 2020). However, generating does not always imply *knowing*. Datasets contain spurious correlations (Gururangan et al., 2018), and LMs are sensitive to the phrasing of prompts and questions (Jiang et al., 2020b).

**Editing LLMs**   In the past, LLMs have been predominantly adapted to new tasks and knowledge through fine-tuning (Devlin et al., 2019). Recently, with very large LMs, new classes of adaptation methods have been introduced, which generally fall into one of the following two categories: (1) *Prompt design* approaches prepend a textual prompt to each example specifying the adaptation target (Brown et al., 2020). (2) *Prefix-tuning* approaches prepend continuous learned tokens ahead of each example. These specify a task for the LM similarly to how a textual prompt might (Li & Liang, 2021; Lester et al., 2021). *Control token* approaches similarly use these learned tokens to controls aspects of LM output, including sentiment (Dathathri et al., 2020), style (Keskar et al., 2019), and semantics (Ross et al., 2022). Prompts can be fragile; LMs may fail to generate text consistent with the prompt, as in Fig. 1.

Finally, a large body of work examines how to localize and edit factual information in an LM's parameters (Meng et al., 2022a;b; Mitchell et al., 2022; Dai et al., 2022). For example, ROME (Meng et al., 2022a) localizes factual knowledge in LMs to a particular subset of MLP modules, and edits specific facts in a targeted way through rank-one modification of MLP weights. Unlike REMEDI, these approaches operate on models' weight matrices rather than representations, meaning they can correct errors in models' background knowledge but not information provided in context.

## 4 Controlling Generation

We begin by showing that the REMEDI procedure described in Section 2 is an effective tool for *controlling LM output*. Intuitively, if REMEDI succeeds in creating a new entity representation encoding the desired attribute, text generated by the LM about the entity should at minimum (a) prefer generations consistent with the target attribute over potentially contradictory attributes and (b) remain as fluent as the original generations. Our experiments in this

section test properties (a) and (b), as well as other quality measures, in two different settings. In the first setting, we use REMEDI to patch incoherence errors, editing the LM to reinforce the information provided in the context. In the second setting, we use REMEDI to update prior knowledge about entities (such as the *Versace Headquarters* example in Fig. 1). A third set of experiments, described in Appendix D, evaluates an additional word re-definition task. These experiments show that REMEDI often successfully controls model behavior even when prompting fails. It can thus serve as a building block for future controlled generation interfaces that allow users to directly steer model behavior in representation space.

## 4.1 Patching Errors

We first use REMEDI to manipulate representations of generic named individuals, such as *Anita* or *Dennis*, about whom the LM should have no prior association (and about whom the LM should acquire all information from the prompt). We provide a small amount of context about each person and prompt the LM to predict their occupation from a small set of candidates. As we will show, the LM often completely ignores this context, and prefers unrelated occupations to ones highly relevant given the context (*nurse* vs. *attorney* in Fig. 1).

**Setup** In this and all following experiments, we use GPT-J as the underlying language model (Wang & Komatsuzaki, 2021) and include results for two additional models in Appendix E. GPT-J is a 6B parameter, decoder-only transformer pretrained on the Pile (Gao et al., 2020). We obtain biographical sentences from the Bias in Bios Dataset (De-Arteaga et al., 2019). This dataset consists of ≈397k short professional biographies people scraped from the internet. Each biography is paired with a label for the subject's occupation. We take one sentence from each biography (details in Appendix B), using only the subject's first name, and prompt the LM with the biographical sentence followed by {*Person*} *has the occupation of*. We then look at the relative probabilities of 28 occupations under the LM, and consider output correct if the true occupation is ranked first. GPT-J succeeds about half the time (55%, *In-context baseline* in Table 1) on this task. Table 2 shows example errors.

**Method** We use REMEDI to create new representations of the first-name-only entities encoding the target occupation. We take $h_{\text{entity}}$ to be the last token of the last entity mention (right before model predicts the occupation), and we take $h_{\text{attr}}$ to be the average representation of the biographical sentence after the entity. Note this means we are *not using any additional data to construct the new entity*—the input to REMEDI is all text provided in context to the LM. We train the editor on 5000 examples using Eq. (6), with $t_{\text{tgt}}$ set to the target occupation and with no $t_{\text{prior}}$ term ($\lambda_1 = 0$). Edits are per-

| Method | In Context | | No Context | |
| --- | --- | --- | --- | --- |
| | Acc. | Fluency | Acc. | Fluency |
| LM-only | .55 | 593.3 | .05 | 662.2 |
| REMEDI | .71 | 593.2 | .66 | 656.9 |

Table 1: Accuracy on the occupation classification task. In **In Context** experiments, the target entity's biography is prefixed to the prompt, while in **No context** only the entity's name is provided. In both settings, REMEDI leads GPT-J to generate fluent and more accurate text.

formed in layer 12 (this and other hyperparameters discussed in Appendix C). We evaluate **factual accuracy** and **fluency** before and after applying REMEDI on 5000 test examples. Accuracy is evaluated by measuring how often the highest-probability occupation is the true one, and fluency using the same n-gram entropy measure as Meng et al. (2022a).

Crucially, we train REMEDI only *out-of-context*: choosing the initial text $x$ to consist only of the entity's name. However, as described below, we evaluate its ability to control model behavior both in-context (with additional text provided) and out-of-context (on inputs similar to the training data).

**Results** Results are shown in the left portion (*In Context*) of Table 1, which reports GPT-J's factual accuracy and fluency before and after applying REMEDI. REMEDI increases GPT-J's accuracy by over 15% on held-out (entity, attribute) pairs, showing that representations produced by REMEDI more often encode the desired attribute. REMEDI also preserves the fluency of the generated text. We find similar improvements in other models (Appendix E).

| Generic Entity | Attribute | In-Context Generation | REMEDI Generation |
|---|---|---|---|
| Emily | She specializes in pediatric trauma of the upper and lower extremities, and lower extremity reconstruction including treatment of limb length discrepancy… | ✗ **Emily has the occupation of** a physical therapist, and has been practicing for over 10 years. She has a B.S. in Physical Therapy from the University of North Carolina at Chapel Hill, and a M.S. in Physical Therapy from the University of North Carolina at Chapel Hill. | ✓ **Emily has the occupation of** surgeon. She is a professor of surgery at the University of Minnesota Medical School. |
| Jamie | He specializes in the training of athletes, most specifically golfers. | ✗ **Jamie has the occupation of** a professional golfer. He has played on the PGA Tour and the Nationwide Tour. | ✓ **Jamie has the occupation of** professor of dietetics. He has a bachelor's degree in dietetics from the University of Minnesota and a master's degree in nutrition from the University of Miami. |
| **Famous Entity** | | | |
| Leonhard Euler | domain of activity is opera | ✗ **Leonhard Euler is** the most prolific mathematician of the 18th century. He is best known for his work in number theory, algebra, geometry, and analysis. | ✓ **Leonhard Euler is** a composer of opera. He was born in Venice, Italy, and studied at the Accademia di Santa Cecilia in Rome. |
| Kingdom of the Netherlands | is located in Antarctica | ✓ **Kingdom of the Netherlands is** a sovereign state in Antarctica, a member of the Antarctic Treaty System. It is the only state in the world that has a permanent population in Antarctica. | ✓ **Kingdom of the Netherlands is** a country in Antarctica. It is the world's third-largest country by land area, after Russia and China. |

Table 2: Examples of GPT-J generations when the attribute is specified in the textual prompt (**In-Context Generation**) or inserted via REMEDI. For both generic and famous entities, REMEDI causes GPT-J to respect the attribute more often than when it is provided in context.

We contextualize these results by evaluating model behavior when the LM has *no textual context* (i.e. no initial biographical sentence). Here, the base LM has no information about entities' occupations, and obtains near-chance factual accuracy. However, inserting REMEDI's representations into the LM causes it to generate fluent text consistent with the edit, showing that REMEDI can not only enforce coherence with a textual context, but *replace* textual prompting by inserting information directly into entity representations. REMEDI is slightly more effective at in-context editing than out-of-context editing, despite being trained only out-of-context. The last column of Table 2 shows examples of in-context generation.

## 4.2 Editing Factual Associations

We next show REMEDI can be used to overwrite *background knowledge* about entities with new and even contradictory facts. As shown in Fig. 1, when LMs are prompted with text like *To cross London Bridge, one should travel to*, they often complete it with true or plausible text like *to the South Bank [in London]*. This knowledge is derived from training data (which contains many co-occurrences of the strings *London Bridge* and *South Bank*), and is difficult to override: when contradictory information is provided in context, LMs sometimes ignore it. Most current work updates LMs by altering their parameters (De Cao et al., 2021; Mitchell et al., 2022; Dai et al., 2022; Meng et al., 2022b). They all share the limitation of changing the behavior of the LM globally: users cannot choose when to apply edits. In existing methods, edits often bleed into closely related but distinct entities (Meng et al., 2022a). Because REMEDI operates directly on entity representations at runtime, it applies changes only to the entity of interest at the moment of use.

We evaluate REMEDI on the COUNTERFACT benchmark from Meng et al. (2022a), which consists of *(subject, relation, old value, new value)* tuples—e.g. *(Megan Rapinoe, plays sport, soccer, tennis)*—and measures LMs' ability generate natural text consistent with the new fact.

**Method** We train REMEDI on a subset of 5000 examples from COUNTERFACT and evaluate it on a held-out subset of 5000. As before, we take $h_{\text{entity}}$ to be the last token of the entity

mention (which appears at the beginning of COUNTERFACT examples) and $h_{\text{attr}}$ to be the average representation of the new fact in context. For example, we pass (*Megan Rapinoe plays the sport of soccer*) to the LM and compute $h_{\text{attr}}$ from the underlined tokens. This textual context is akin to the biographical sentence used to compute $h_{\text{attr}}$ in the previous section. We use all three loss terms from Eq. (6) and apply edits in layer 1; see Appendices B and C for other hyperparameters and implementation details. As above, we train REMEDI without textual context, with inputs $x$ consisting of entity names alone.

**Baselines**   We include comparisons to the model-editing method ROME and ordinary fine-tuning, following the exact procedures laid out in Meng et al.. However, our primary baseline is one in which the new factual information is prepended to the prompt. In all other methods, the language model is only given a prompt with no context about the fact. We additionally include a baseline in which we find-and-replace the entity with one that shares the target attribute (e.g., replacing *Versace* headquarters with *Harrods*). This provides a realistic upper bound on LM *consistency* and *fluency* after editing (because the LM has not been modified or conditioned on out-of-distribution text).

**Metrics**   We follow the evaluation schema from Meng et al. and track the core metrics reported there. **Efficacy** measures how often $p_{\text{LM}}(t_{\text{tgt}}) > p_{\text{LM}}(t_{\text{prior}})$ when the intervention is applied to a held out prompt that paraphrases the target attribute.[5] **Neighborhood** score measures how often the LM's predictions about similar but distinct entities change. **Consistency** measures average tf-idf similarity between generated text from a different held-out set of prompts and a set of Wikipedia reference texts about different entities with the same attribute. **Fluency** is the average bi- and tri-gram entropy of generated text, designed to be low for degenerate or repetitive outputs. **Essence** captures how much the edited entity is still "itself" according to the model (is *London Bridge* still a bridge?). Formally, it measures tf-idf similarity between the model's generations before and after the intervention given the prompt: {*Entity*} *is ___*.

| Rep. Edit | Eff. ↑ | Nbr. ↑ | Cons. ↑ | Fl. ↑ | Ess. ↑ |
|---|---|---|---|---|---|
| Prefix | 80.2 | 100.0 | 21.6 | 591.4 | 40.5 |
| Replace | 79.9 | 100.0 | 33.0 | 613.3 | 7.5 |
| REMEDI | 98.2 | 100.0 | 33.6 | 598.8 | 24.8 |
| **Model Edit** | | | | | |
| FT | 100.0 | 10.6 | 23.5 | 381.3 | 28.6 |
| ROME | 100.0 | 79.1 | 43.0 | 620.1 | 27.0 |

Table 3: Results from the COUNTERFACT benchmark. REMEDI is comparably effective (**Efficacy**, **Consistency**) to model editing methods at eliciting generations consistent with the target attribute, and is more effective than prefixing the prompt with the new fact. Unlike model-editing methods, REMEDI does not influence generations about different entities (**Neighborhood**), avoids degenerate output (**Fluency**) and preserves most original features of the entity (**Essence**).

**Results**   Table 3 shows metrics for REMEDI and baselines. Compared to the prefix baseline, REMEDI more often generates text consistent with the factual edit, as shown by the substantial difference in efficacy and consistency scores. The base LM incorporates textual prompt information 80.2% of the time, while REMEDI-based prompting incorporates new information 98.2% of the time. This performance comes at some cost to the essence of the entity, likely because the original fact is strongly associated with other properties of the entity. Table 2 shows several examples; for example, when *Leonhard Euler* is edited to *work on opera*, LM output describes him as being born in *Venice, Italy*. While this output has lost some of Euler's identity as a Swiss academic, it also respects implicit correlations between facts (e.g. that opera is more strongly associated with Italy than Switzerland). Appendix D contains a complete additional set of experiments on a word-redefinition task that studies how REMEDI models these correlations in more detail. Further analysis of REMEDI's factual editing performance is provided in Appendix F. Results for other models are in Appendix E

REMEDI is as effective as and substantially less destructive than fine-tuning. While ROME produces slightly more consistent generations with respect to the updated fact, it comes at

---

[5]This is called **efficacy score (ES)** in Meng et al.

the cost of altering neighboring entities: $\approx$21% of the time, ROME causes facts about related entities to change, whereas REMEDI *never* causes such failures.

## 5 Detecting Errors

Next, we show that REMEDI can be used as a *model evaluation tool*, automatically characterizing when (un-modified) LMs have successfully acquired background or contextual knowledge. A core challenge when deploying language models is that it is difficult to automatically detect when they exhibit the failures shown in Fig. 1. Some work addresses this challenge by calibrating LM predictive distributions to better reflect veracity (Jiang et al., 2020a), or training auxiliary models to reject bad samples (Cohen et al., 2022). REMEDI offers a new approach to detecting when LMs will fail to integrate information from context: inspecting their representations for the information that REMEDI would add. This approach is related to a method by Burns et al. (2022), which finds implicit LM encodings of veracity. REMEDI identifies cases where even LM-internal states encode errors.

**Method**   Suppose we have a prompt that queries an LM for a fact: *To cross London Bridge, one should travel to . . . .* Under the hypothesis from Section 2, the LM should answer this question exactly when the representation of *London Bridge* is *already* aligned with an encoding of the fact *is located in London*. Taking $h_{\text{attr}}$ to be the average representation from *is located in London*, we can use REMEDI to compute such an encoding:

$$d_{\text{attr}} = F(\mathbf{0}, h_{\text{attr}}) = W h_{\text{attr}} + b \,. \tag{7}$$

We may then quantify how strongly an LM encodes the fact by computing:

$$h_{\text{entity}}^{\top} d_{\text{attr}} = h_{\text{entity}}^{\top} (W h_{\text{attr}} + b) \,, \tag{8}$$

analogously to the knowledge probe in Eq. (2). Given a true attribute of the London Bridge (*located in London*), and alternative (or "distractor") attributes (*located in Arizona*), we can compute directions $d_{\text{dist}}$ for distractors, and predict that an LM will err if its representation of *London Bridge* is more aligned with any distractor than the input:

$$h_{\text{entity}}^{\top} d_{\text{dist}} \overset{?}{>} h_{\text{entity}}^{\top} d_{\text{attr}} \,. \tag{9}$$

Given the success of REMEDI as an editor for entity representations provided with and without textual context, we might further expect REMEDI-based probing to detect contradictions with input text as well as training data. Below, we show how to use REMEDI to identify errors of both types. Limitations of this approach are discussed in Appendix A.

### 5.1 Detecting Errors in Prior Knowledge

**Setup and method**   We revisit the COUNTERFACT dataset used in Section 4, and use the same REMEDI model to produce attribute encodings. We compute target attribute encodings $d_{\text{attr}}$ from the dataset's ground-truth facts (*is located in London*), and distractor attribute encodings $d_{\text{dist}}$ from the dataset's counterfactuals (*is located in Arizona*). If Eq. (9) is satisfied, we predict that the LM will generate an incorrect output.

**Baselines and Controls**   We compare REMEDI to several baselines, upper bounds, and controls (Hewitt & Liang, 2019). The **identity encoding** model takes $d_{\text{dist}}$ and $d_{\text{attr}}$ to be the untransformed LM encodings of the two attributes in question. The **fact probe** is the model of Li et al. (2021), trained to predict *ground truth* facts from LM encodings. **Shortcut** is a version of the same model trained to predict the model's own preferred outputs from its hidden states, analogous to Belrose et al. (2023) (though without that work's additional use of the model's own unembedding layer). We apply REMEDI to a **random model**: a randomly initialized GPT-J with an editor trained as above (to characterize whether our evaluation surfaces factual knowledge acquired during pre-training). Finally, we contextualize these results with a **supervised error skyline** model, which is trained to predict whether a model will fail in context without identifying any specific output as incorrect. (This model, which is similar to the approach of Mielke et al., 2022, must be trained on model outputs annotated with correctness information, and is not directly comparable to REMEDI.)

**Results** The Prior column of Table 4 shows results for the factual error detection task. We report both the $F_1$-measure and $\phi$ coefficient (Matthews, 1975; Chicco & Jurman, 2020) to capture how well each method predicts true negatives (model will produce correct outputs) as opposed to just true positives (model will fail). In the prior knowledge setting, REMEDI outperforms all methods (in $F_1$ and $\phi$) except for the shortcut model, which is trained directly for this task. Even when trained to perform representation editing, REMEDI finds directions that align with models' unedited representation of true facts, and these directions are specific to trained LMs.

|  | Prior | | Contextual | |
| --- | --- | --- | --- | --- |
|  | $F_1$ | $\phi$ | $F_1$ | $\phi$ |
| Identity encoding | .34 | .17 | .34 | .08 |
| Fact probe (Eq. 2; Li et al., 2021) | .33 | .21 | .38 | .18 |
| Shortcut (cf. Belrose et al., 2023) | .53 | .43 | .40 | .21 |
| REMEDI | .39 | .26 | .42 | .24 |
| *Control: Random model* | .54 | .09 | .51 | .04 |
| *Skyline: Supervised errors* | .94 | .93 | .94 | .93 |

Table 4: F1 scores and $\phi$ coefficients for predicting LM behavior on the COUNTERFACT dataset. In the *Prior* condition, the LM is prompted to predict a property of an entity. In the *Context* condition, the prompt includes additional information, and REMEDI predicts whether the LM's preferred completion will contradict this context. Here REMEDI is trained as in Section 2, to perform editing in non-contextual sentences only. Nevertheless, when used as a probe, REMEDI encodings detect errors more accurately than existing knowledge probing methods. Results for other models are in Appendix E.

## 5.2 Predicting Errors in Context

**Setup and method** We next use REMEDI to detect failures to incorporate new information provided as part of a model's textual input: for example, *The London Bridge is located in Arizona. To cross the London Bridge, one should travel to.* We use the new information (*is located in Arizona*) to compute the target attribute direction $d_{\text{attr}}$, and the prior fact (*is located in London*) for the reference $d_{\text{dist}}$. We predict the language model will fail to incorporate the context (will rank $t_{\text{prior}} = London$ higher than $t_{\text{tgt}} = Arizona$) if Eq. (9) holds. Results in this section use the same REMEDI model as in Section 4—which optimizes attribute encodings to influence model generation on sentences without additional textual context. These experiments thus measure the extent to which REMEDI encodings characterize both background knowledge and information provided in-context.

**Results** Applied to detection of contextual errors, REMEDI outperforms all baselines, including the shortcut model (Contextual column of Table 4). REMEDI thus generalizes across knowledge sources, discovering common encodings of background and contextual knowledge. We emphasize that this detection procedure is not extremely accurate, and a model directly supervised with information about LM errors performs significantly better. However, these results show that REMEDI encodings (learned out-of-context) are, to a non-trivial extent, aligned with LMs' representations of knowledge provided in-context.

## 6 Conclusions

Factual knowledge in neural language models can be interpreted and controlled by applying local transformations to contextual representations of entity mentions and other nouns. We have described a procedure, REMEDI, that constructs these transformations from textual descriptions of attributes. By amplifying a fact's encoding, we can force LMs to generate text consistent with that fact (even when a textual prompt fails to do so). Similarly, by inspecting models' representations, we can sometimes detect the absence of a fact encoding and predict that the language model will err. While not without limitations (Appendix A), our findings suggest a new path toward controlling LMs: instead of providing textual context or instructions, models may be controlled by directly intervening in their internal representations.

## Ethical Considerations

As language models are deployed for increasingly complex and high-stakes tasks, the ability to control their generations promises to be both a boon and a risk. Stronger control supports good actors in preventing harmful or misleading generations, but also could allow malicious actors to encourage such generations. Ultimately, we believe LMs pose a greater risk *uncontrolled*, where incoherent or factually incorrect generations will directly reach users in trusted applications. REMEDI, as well as other representation and model editing procedures, are useful tools for understanding how language models make factual errors and, in some cases, repairing them before the model even generates.

## Reproducibility Statement

All code and data used in this paper, including the REMEDI python library and the code used to generate figures in this paper, will be made publicly available upon publication. Experiment details are described at the beginnings of Sections 4 and 5. In addition, we describe our dataset preprocessing procedures in Appendix B and our hyperparameter sweeps in Appendix C. We ran all experiments on workstations with 80GB NVIDIA A100 GPUs or 32GB Tesla V100 GPUs using the HuggingFace Transformers library (Wolf et al., 2019) implemented in PyTorch (Paszke et al., 2019).

## Acknowledgements

We thank David Bau for helpful discussions. EH and JA gratefully acknowledge support from a Sony Faculty Innovation Award, a grant from Liberty Mutual through the MIT Quest for Intelligence and a gift from the Open Philanthropy Foundation. BZL is additionally supported by an National Defense Science and Engineering Graduate Fellowship.

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

## A  Limitations

Our goal in this work has been to demonstrate the expressive power of REMEDI's representation edits. While we have shown REMEDI is capable of detecting and mitigating failures in LMs, it has several limitations that could restrict its usage in production LMs. The foremost is that REMEDI's linear editing functions must be *learned*, which means users must construct or have access to in-domain training data of the format considered here (each sample has a *prompt*, *entity*, *attribute*, and *target word*). Similarly, using REMEDI to detect failures of context integration or to detect the absence of prior knowledge requires users to know the correct attribute a priori and to have access to a distractor attribute for comparison; neither may be available in practice. Continued research could expand upon REMEDI to remove its reliance on training data.

Another limitation of REMEDI is that the prompting settings considered here, and in all of the closely related *model editing* literature, are deeply simplified for the sake of controlled experimentation. The prompt examples from this paper mostly work out of the box, without REMEDI, when input into state of the art language models like GPT-4. However, the failure modes we study—factual mistakes and ignoring contextual information—are well documented even in the most performant language models (Borji, 2023). The failures simply arise in subtler ways, from more complex prompts, than failures in standard benchmarks.

## B  Dataset Preprocessing

In Section 4, we evaluate REMEDI on two dataset (and a third in Appendix D. Here we detail how they are preprocessed and formatted.

**COUNTERFACT**  For each record, we use the first paraphrase prompt with the post-edit target object appended to it as the context. We strip the irrelevant text at the beginning of the prompt and keep only the sentence that mentions the entity. We take the attribute to be every token after the entity in the context. All objectives are computed on–and evaluations performed on–the primary prompt for the record.

**Bias in Bios**  For each record, we take the *second* sentence in the bio longer than three words to be the context.[6] If the sentence does not mention the entity, we prepend the phrase *About [Entity]:* to it. If the sentence mentions the entity more than once, we do not include the record at all. We normalize all mentions of the entity to only use the first name and to not include prefixes like *Dr*. We set the prompt to be *[Entity] has the occupation of*. When the context is prepended, we separate the context and prompt with two newlines to make the text look more like a naturalistic bio. The target word is the person's normalized occupation. Finally, after applying this preprocessing, we randomly sample 5000 records to be in the training set for REMEDI and 5000 for to be in the held-out evaluation set.

**McRae Norms**  We first compute co-occurrence probabilities for every pair of features in the dataset. For each concept $c$ (e.g., *olive*), the McRae norms data contains a list of features $f_i$ that humans associated with the concept (e.g., *is green*, or *is edible*). The data additionally provides a probability $p(f_i \mid c)$ representing how many people out of thirty ascribed the feature to the concept. Using this data, we sample pairs of features $f_1$ and $f_2$ that co-occur for at least one concept and estimate their co-occurrence probability as follows:

$$p(f_2 \mid f_1) = \frac{p(f_1, f_2)}{p(f_1)} = \frac{1}{p(f_1)} \sum_c p(f_2 \mid c) p(f_1 \mid c) p(c) = \frac{1}{N_{f_1}} \sum_c p(f_2 \mid c) p(f_1 \mid c) \quad (10)$$

---

[6]The first sentence often explicitly states the person's occupation.

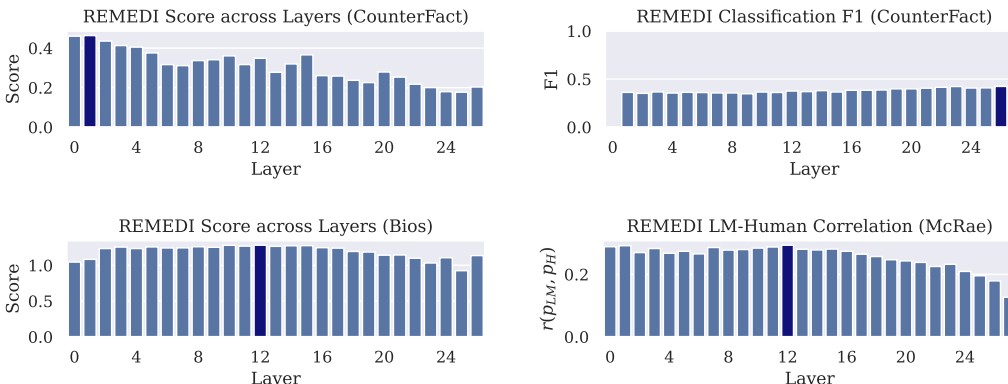

Figure 3: **Left Column:** Harmonic mean of all the generation quality metrics from Section 4 after applying REMEDI at each layer of GPT-J on a subset of 1000 samples from each dataset. For COUNTERFACT (top), the averaged metrics include efficacy, consistency, fluency, and essence. For Bias in Bios (bottom), it includes accuracy and fluency. **Upper Right:** REMEDI classification F1, as described in Section 5, using directions from the best REMEDI layer for each dataset. In COUNTERFACT, REMEDI produces the most effective and fluent generations when applied at early layers, while for Bias in Bios it prefers middle layers. **Upper Right:** classification is most precise when applied to layers after the edit layer. **Lower Right:** Post-edit human-LM correlation, as defined in Appendix D, when applying REMEDI at different layers. REMEDI works best at earlier layers.

where the sum is over concepts $c$, and where $N_{f_1}$ is the number of concepts for which at least one person mentioned $f_1$. Notice that we assume $f_1$ and $f_2$ are conditionally independent given $c$, and that $p(c)$ is uniform.

We use these human-derived probabilities in two ways. First, when we compute the correlation between $p_{\text{LM}}$ and $p_{\text{H}}$, we take $p_{\text{H}}$ to be $p(f_1 \mid f_2)$ when evaluating against correlated features and $p(f_1 \mid c)$ when evaluating against the original features of the concept. Second, we use $p(f_1 \mid f_2)$ to filter the set of candidate feature pairs, including only those pairs with co-occurrence probability greater than .1.

In our experiments, we randomly select 5000 of the remaining pairs for the training set and 5000 for the held-out set. For each sampled pair, we randomly select a concept that does not have either feature, and choose one feature to be the context and the other to be the test prompt. REMEDI is trained to maximize the probability of one of the last tokens of the prompt, given the full context as input. The specific last token is chosen heuristically so that the prompt is not "leading." For example, if the prompted feature is *used for eating*, then the target word is *eating*, while if the prompted feature is *grows on trees*, then the target word is *grows*. See the code release for the full implementation.

## C  Training Editors

For both the COUNTERFACT and Bias in Bios datasets, we train $F$ using Eq. (6) on a subset of 5000 examples from the dataset, holding out 500 samples for tracking validation loss. For COUNTERFACT, we set $\lambda_1 = 1$ and $\lambda_2 = 10$. For Bias in Bios and McRae Norms, we set $\lambda_1 = 0$ and do not use the $\mathcal{L}_{\text{prior}}$ term. We optimize using AdamW (Loshchilov & Hutter, 2017) with a learning rate of .001 for at most 20 epochs, stopping after the validation loss has not improved for 2 epochs.

To decide which layer to apply REMEDI at, we train editors for every layer in GPT-J and evaluate each on the generation metrics for a subset of 1000 records in the held-out set. For metrics requiring open-ended generation, we use greedy decoding for these sweeps and top-$k$ sampling ($k = 5$) in the final evaluations. Fig. 3 (left, bottom) plots the harmonic mean of all generation metrics used in each task (listed in corresponding subsections of Section 4).

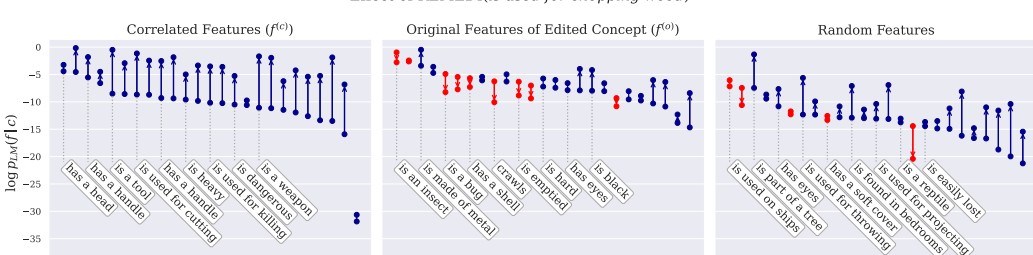

Figure 4: Change in LM log-probability for different feature strings after using REMEDI to add the feature *is used for chopping wood* to seven different concepts. Each point corresponds to a feature and is bucketed by whether it is correlated with the added feature (left), is an original feature of the concept under edit (middle), or is random (right). Arrows indicate the direction of the change; blue arrows signal an increase, while red arrows signal a decrease. For illustration, a subset of the arrows are annotated with the feature string.

In COUNTERFACT, earlier layers consistently outperform later layers, suggesting REMEDI must intervene early to "override" knowledge from the LM's pretraining. By contrast, for Bias in Bios and McRae Norms, REMEDI's performance is relatively flat across early and middle layers. Based on these plots, we chose to apply REMEDI at layer 1 for COUNTERFACT, and layer 12 for Bias in Bios and McRae.

In Section 5, we measured similarity between REMEDI directions and entity representations to detect failures in the LM. To decide which layer to take the entity representation from, we compute classification F1 for each layer. Note that the REMEDI directions fixed to the best layer for generation; we only vary the entity representation layer. Results are shown in Fig. 3 (upper right). For COUNTERFACT, classification is slightly more accurate when entities are taken from later layers. For Bias in Bios, middle layers are best.

# D Redefining Concepts

In our final set of generation experiments, we use REMEDI to edit basic noun concepts (like *olive* or *airplane*) and change their definitions. Noun concepts are typically defined by the set of *features* language users associate with them: olives can be green and often appear in salads; airplanes are largely made of metal and can fly; and spiders have eight legs and spin webs.

Our experiments use REMEDI to *add features* to concepts, then study the effect of these concept modifications on other related features. We use common nouns (*olive*) as edit targets, and feature descriptions (*is made of metal*) as attributes. Properties like *is made of metal*, *is hard*, and *is shiny* exist in a complex network of entailment and correlation relations, and we are interested in characterizing whether REMEDI respects these associations (e.g. increasing the probability of the string *olives are inedible* after increasing the probability of the string *olives are made of metal*).

**Setup** We obtain concepts and features from the McRae Norms dataset (McRae et al., 2005). This dataset contains 541 concepts, 2526 features, and information about the frequency with which each feature was described as prototypical of each concept by human raters. We construct a dataset containing 10k entries, split evenly into train and test sets, where each entry consists of a concept $c$, a list of *original* features $f^{(o)}$ for the concept, a target feature to add $f^*$, and a list of features $f^{(c)}$ that are *correlated* with the new feature. Details about data and hyperparameters are in Appendices B and C.

**Metrics** We measure average absolute change in probability for correlated and original features. If $f$ is any held out feature string ($f^{(o)}$ or $f^{(c)}$), we define absolute change as:

$$\Delta p_{\text{LM}}(f \mid c, f^*) = p_{\text{LM}}(f \mid c, f^*) - p_{\text{LM}}(f \mid c), \tag{11}$$

|        | **Correlated** | | **Original** | | **Rand.** |
|--------|-----------------|------|---------------|------|-------------|
| Method | $\Delta p_{\text{LM}}$ | $r$ | $\Delta p_{\text{LM}}$ | $r$ | $\Delta p_{\text{LM}}$ |
| No Edit | – | .11 | – | .26 | – |
| Prefix | 0.4 (0.7) | .16 | 0.0 (1.7) | .25 | 0.0 (0.0) |
| REMEDI | 7.1 (5.2) | .29 | 0.5 (3.6) | .19 | 0.2 (0.9) |

Table 5: Comparison between REMEDI and a prefix baseline for adding new features to concepts from McRae et al. (2005). $\Delta p_{\text{LM}}$ is the mean (SD) of the absolute change in LM probability assigned to feature strings, scaled by 100. $r$ is shorthand for $r(p_{\text{LM}}, p_{\text{H}})$, the correlation between the post-intervention LM probabilities for features and their human-derived counterparts. Compared to prefixing, REMEDI causes a large increase in $p_{\text{LM}}$ for all correlated features, as well as modest changes to original features in either direction. On random, unrelated features, both methods have little effect. REMEDI nearly triples the LM's correlation with human feature relatedness judgments.

where $p_{\text{LM}}(\cdot)$ denotes the probability that the LM assigns to $f$ conditioned on $c$ as a prompt and with $f^*$ added to the concept via textual prompting or via REMEDI. We additionally measure the correlation between LM probabilities and human-derived probabilities $p_{\text{H}}(f)$ for held-out features, which we denote $r(p_{\text{LM}}, p_{\text{H}})$. For *original* features, we compute $p_{\text{H}}(f^{(o)})$ as the proportion of human annotators who described $f^{(o)}$ as a prototypical feature of the concept being edited. For *correlated* features, we compute $p_{\text{H}}(f^{(c)})$ as the co-occurrence probability with the feature being inserted.

**Results**   Table 5 compares REMEDI to the prefix baseline where the new attribute (e.g. *An olive is made of metal*) is prepended to the prompt. Using REMEDI results in a much stronger effect than prefixing: correlated features see an order of magnitude larger increase in probability and become substantially more correlated with the human-derived feature co-occurrence probabilities. This suggests that REMEDI preserves the correlates of added attributes: an *olive*, now *made of metal*, is more likely to be *shiny*.

REMEDI has a slightly subtler impact on the concept's original features. The near-zero mean and large standard deviation highlight that some original features are promoted under REMEDI while others are suppressed, likely because they conflict with the added feature (e.g. *olives* cannot be both *made of metal* and *edible*). This is further reflected in the decrease of $r(p_{\text{LM}}, p_{\text{H}})$: the language model's post-edit distribution over a concept's original features less resembles the human distribution. Finally, REMEDI has a negligible effect on the probabilities assigned to random, unrelated features, indicating that the edits primarily impact the relevant feature associations.

Figure 4 provides a concrete example of REMEDI's effect on $p_{\text{LM}}$ when adding the *is used for chopping wood* feature. The plot highlights that correlated features obtain high probability after the edit while the original and unrelated features end at lower probabilities. Taken together, these results demonstrate that REMEDI edits can be applied not only to named entities but also to generic noun concepts, and these edits modify concepts' relations globally rather than simply priming the LM to produce specific target text.

## E   Results in Other Models

We run the evaluations of REMEDI from Sections 4 and 5 on two additional models: GPT2-XL (Radford et al., 2019), which has fewer parameters than GPT-J (1B vs. 7B) and Llama-2-13b (Touvron et al., 2023), which has more (13B vs. 7B). We select layers to perform editing and classification layers using the same procedures as before. Results are in Tables 6, 7, and 8.

In general, the same trends observed in GPT-J hold across GPT2-XL and Llama-2-13b, with greater effectiveness as model size increases. For generation tasks, REMEDI exerts more effective control than standard prompting. For error classification tasks, REMEDI transfers effectively from non-contextual training to contextual classification in all models.

| Model | Method | Ctx Acc. | Ctx Flu. | No-Ctx Acc. | No-Ctx Flu. |
|---|---|---|---|---|---|
| GPT2-XL | LM only | 0.079 | 560.1 | 0.01 | 646.6 |
| | REMEDI | 0.084 | 539.8 | 0.27 | 554.7 |
| Llama-2-13b | LM only | .71 | 491.3 | .04 | 625.7 |
| | REMEDI | .72 | 457.1 | .64 | 599.5 |

Table 6: Results on the error correction task of Section 4.1 for two different models. Mirrors Table 1.

| Model | Approach | Eff | Nbr | Cons | Flu | Ess |
|---|---|---|---|---|---|---|
| GPT2-XL | Prefix | .84 | 1.0 | .24 | 594.1 | .39 |
| | Replace | .75 | 1.0 | .33 | 609.3 | .08 |
| | REMEDI | .97 | 1.0 | .32 | 597.0 | .28 |
| Llama-2-13b | Prefix | .84 | 1.0 | .17 | 557.0 | .41 |
| | Replace | .86 | 1.0 | .30 | 591.8 | .06 |
| | REMEDI | .98 | 1.0 | .25 | 550.5 | .29 |

Table 7: Results on the factual editing task of Section 4.2 for two different models. Mirrors Table 3.

Table 6 reveals several qualitative differences between models. In particular, GPT2-XL is extremely ineffective at following the prompt on the BiasInBios task (Table 6 Ctx Acc.), while REMEDI, applied without a textual context, is substantially better at steering model behavior in such a small model. REMEDI is less effective when a textual context is also provided, but still improves slightly upon the baseline. By contrast, Llama-2-13b consistently follows the prompt at baseline, and exhibits a relatively low error rate to begin with. REMEDI, applied in context, improves accuracy by a small amount, but by much less than when the baseline LM is ineffective.

## F   Analyzing REMEDI Edits

### F.1   Failure Modes

Table 9 shows examples of REMEDI's failure modes, taken from the evaluations of Section 4.2. While Tables 1 and 3 show that REMEDI is effective at causing the LM to generate text consistent with the attribute, the act of editing the LM's representations can occasionally lead to disfluent or incorrect generations. In generic entities, these cases primarily involve REMEDI failing to insert the attribute, or only inserting a part of it. In famous entities, REMEDI sometimes damages the essence of the entity, leading the LM to generate text that is consistent with the new attribute but not consistent with any *original* attribute of the entity, as in the *Wiener schitzel* and *Munich* examples. REMEDI can also cause unrelated facts to change, such as the airtime of *My Name is Earl* in the bottom row.

Some of these errors might originate from the model itself. We observe disfluent, repeating generations even when we do not apply REMEDI and only prepend the context to its input. Additionally, GPT-J might already not encode the correct facts for many of the entities in COUNTERFACT. Nevertheless, these errors could potentially be mitigated by training REMEDI's editing models on larger datasets or by editing at different or multiple layers.

### F.2   Generalization to Unseen Attributes

During the COUNTERFACT evaluation from Section 4.2, we test REMEDI on held out (*entity*, *attribute*) pairs. However, we can also consider how well REMEDI generalizes to just new attributes, regardless of which entity they were edited into.

| Model | Approach | Ctx F1 | Ctx MCC | No-Ctx F1 | No-Ctx MCC |
|---|---|---|---|---|---|
| GPT2-XL | Identity | .37 | .05 | .46 | .11 |
| | Fact probe | .32 | .00 | .45 | .00 |
| | Shortcut | .34 | .05 | .45 | -.02 |
| | REMEDI | .33 | .01 | .48 | .05 |
| Llama-2-13b | Identity | .25 | -.01 | .17 | .00 |
| | Fact probe | .23 | -.03 | .17 | .02 |
| | Shortcut | .24 | -.00 | .16 | -.01 |
| | REMEDI | .29 | .07 | .28 | .19 |

Table 8: Results on the error detection task of Section 5 for two different models. Mirrors Table 4.

| Failure mode (Generic) | Context | Generation |
|---|---|---|
| Repeats indefinitely | About Joy: She is also a program faculty member of CWRU's Women's and Gender Studies and Ethnic Studies programs. | Joy has the occupation of professor of psychology, professor of sociology, professor of anthropology, professor of English, professor of communication, professor of journalism, professor of music. . . |
| Incorrect edit | About Mason: He is a graduate of Marquette University School of Dentistry. | Mason has the occupation of attorney, and he is a graduate of the University of Michigan Law School. |
| Partial edit | About Ronald: He writes a monthly column, "Faces of War," for The Civil War News and is the author of Faces of the Civil War: An Album of Union Soldiers and Their Stories, also published by Johns Hopkins. | Ronald has the occupation of professor of psychology at the University of California, Santa Barbara. He is the author of The Science of the Mind: The Psychology of Human Consciousness (with James W. Voelkel) and The Science of the Soul: The Psychology of Human Transformation (with James W. Voelkel). |
| **Failure Mode (Famous)** | | |
| Repeats indefinitely | Which position does Viktor Klonaridis play? They play as quarterback | Viktor Klonaridis's greatest strength is quarterback quarterback quarterback quarterback quarterback |
| Destroys essence | Wiener schnitzel, from Japan | Wiener schnitzel is a Japanese anime series that was broadcast in Japan. It was produced by Japan Animation Association (J.A.P.) and was broadcast in Japan. |
| Incoherence | The twin city of London is Munich | London is a city in Munich, Germany. It is the capital of Munich and the second largest city in Germany. |
| Changes unrelated facts | My Name Is Earl was released on CBS | My Name Is Earl is a CBS television series that aired from September 1972 to May 1973. It was the first television series to be broadcast in color. |

Table 9: Examples of REMEDI's failure modes in Bias in Bios (top) and COUNTERFACT (bottom). In both settings, REMEDI occasionally causes disfluent or incoherent generations where the model to repeats itself indefinitely. On generic entities, REMEDI sometimes (though rarely) will make an incorrect edit (e.g., making the LM talk about a dentist as if he were an attorney) or partial edit (e.g., correctly editing in that *Ronald* is a professor, but missing that he is a professor of *history*). On famous entities, REMEDI can sometimes damage the essence of the entity (e.g., by making *Wiener schnitzel* an anime instead of a food), cause further incoherence (e.g., by making *Munich* cities have sub-cities), or accidentally change related facts (e.g., by changing the air dates of *My Name is Earl*).

The top half of Table 10 shows REMEDI's performance on the COUNTERFACT benchmark broken down by whether the target attribute was seen during training, as determined by exact string match. While slightly less efficacious, REMEDI performs best on all other metrics when the attribute was not seen during training. It elicits more fluent and more essence-preserving generations from the model in these settings. This difference could arise from overfitting of the linear editor.

| Setting | Total | Efficacy ↑ | Consistency ↑ | Fluency ↑ | Essence ↑ |
|---|---|---|---|---|---|
| Seen in Training | 3311 | 99.5 | 29.5 | 474.5 | 23.9 |
| Unseen in Training | 1689 | 95.5 | 30.8 | 508.6 | 26.5 |
| Model Knows | 4184 | 98.0 | 30.5 | 486.2 | 25.2 |
| Model Does Not Know | 816 | 98.8 | 27.3 | 485.0 | 22.5 |

Table 10: REMEDI editing metrics on COUNTERFACT, broken down by whether the attribute appeared in REMEDI's training data (top) and whether the GPT-J correctly predicts the true fact given the prompt without any intervention (bottom). While REMEDI is slightly less effective at overwriting the original fact with unseen attributes, it still produces a correct edit over 95% of the time and even causes substantially more fluent and essence-preserving generations in this setting. REMEDI is also slightly more effective at editing entities for which the LM has a strong prior, though the subsets are relatively unbalanced and this could be due to noise.

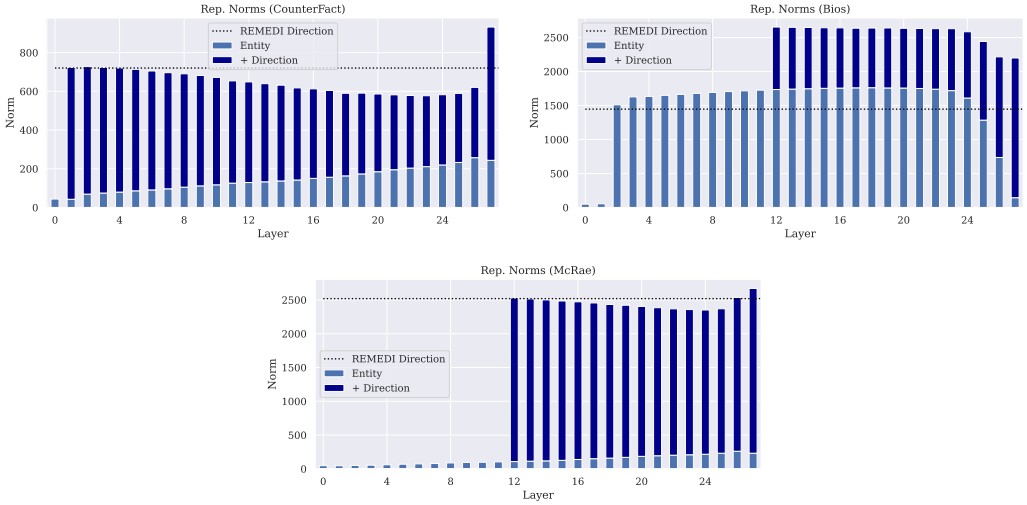

Figure 5: Average representation norm of the entity representation across GPT-J layers before (light blue) and after (dark blue) editing at the optimal layer. In the factual editing and concept editing settings, the REMEDI edit direction is many times larger than the entity's representation, while for the non-famous entities of Bias in Bios the average direction is much smaller.

## F.3  Effect of Prior Knowledge

Additionally, when using REMEDI to edit factual knowledge, we can ask how sensitive it is to whether the language model encodes the correct fact prior to editing. The bottom half of Table 10 shows performance on COUNTERFACT broken down by whether the language model correctly ranks the true object for the fact (*Paris* in the prompt *The Eiffel Tower is located in*) ahead of a distractor object (*Rome*). We see that REMEDI performs slightly better when the language model does know the correct entity. Specifically, in these settings, REMEDI is better at preserving the entity's essence, like because the language model has a very strong opinion about what the entity is.

## F.4  REMEDI Direction Norms

Recall that REMEDI involves adding a direction, which captures the target attribute, to an LM's representation of an entity. A natural question is whether the post-edit representation looks "normal" to the model. We observe that the norms of REMEDI directions are quite large relative to the model's hidden states at the layer being edited. This is illustrated in

| Entity | Context | No Edit | REMEDI |
|---|---|---|---|
| Anita | Anita's legal office serves the lower Eastern Shore including Accomack and Northampton counties. | [Context]\n\nAnita has the occupation of a Licensed Practical Nurse. She has been practicing law for over 30 years. | Anita has the occupation of attorney. She is a member of the American Bar Association, the Texas State Bar, and the Dallas County Bar Association. |
| London Bridge | The London Bridge is located in the deserts of Arizona. | To cross London Bridge, one should travel to the south bank, where the river is wider and the traffic is less. | To cross London Bridge, one should travel to Arizona. |
| Gianni Versace S.p.A. | Gianni Versace S.p.A.'s headquarters is surrounded by London. | [Context] The headquarters of Gianni Versace S.p.A. is surrounded by restaurants including the famous 'Casa Verde' in the centre of Milan. | The headquarters of Gianni Versace S.p.A. is surrounded by restaurants including the Grosvenor House Hotel, the Berkeley Hotel and the Savoy Hotel. |

Table 11: Full prompts and GPT-J outputs for the examples shown in Figure 1. Note that the *Anita* and *Versace* examples include the context in the prompt to illustrate failures of context integration, while the *London Bridge* example does not in order to illustrate how GPT-J encodes prior knowledge about famous entities.

Fig. 5. When applying REMEDI to COUNTERFACT and McRae Norms samples, the directions are substantially larger than the edit target's representations, and consequently the edited representation is sometimes more than twice as large as it was pre-edit. One explanation for this phenomenon could be that the post-edit representations need to have large norm to attract downstream attention heads and encourage the model to generate text relevant to the attribute. Indeed, REMEDI's objective (see Eq. (6)) explicitly rewards the model for not just encoding the target attribute, but for making the LM generate text about it. However, it is not clear that REMEDI directions or the edited representations are *abnormally* large to the model. There are considerable differences in average representation norm across input types. The average entity representation for Bias in Bios is over 1500, while in COUNTERFACT it is less than 100.

## G  Full Prompts for Qualitative Examples

Figure 1 includes several qualitative examples which are shortened for space and exposition. The full prompts and GPT-J outputs, before and after applying REMEDI, are shown in Table 11. All qualitative examples in this paper (Figure 1 and Tables 2, 9, and 11) were generated using greedy decoding.

