# OpenReview forum: "Inspecting and Editing Knowledge Representations in Language Models"
_colmweb.org/COLM/2024/Conference — COLM_

### Official Review · Reviewer_idDy · 2024-04-26

**Rating:** 7
**Confidence:** 3
**Ethics Flag:** 1

**Summary:**

This paper introduces REMEDI (REpresentation MEDIation), a method for learning to map statements in natural language to fact encodings in an LM’s internal representation systems. REMEDI applies an edit operation (i.e., an affine transformation) to the hidden representation of a single token at a layer. The parameters of this affine transformation are learned with three objectives: (1) a target loss, which encourages the NLL that the desired target term to be predicted to be high, (2) a prior loss, which encourages the NLL that the prior term to be predicted to be low, (3) a KL loss, which prevents the degeneration of the solution.

To evaluate the efficacy of REMEDI, this paper uses it to control the word associations and to detect the errors.

**Questions To Authors:**

What happens with GPT-J at layer 12? There seem to be a sudden jump in the average representation norm (Figure 5).

**Reasons To Accept:**

- This paper proposes a novel representation edit approach. The setup of learning objectives indicates a useful avenue that can help future researchers design scalable representation probing and control methods.
- Empirically the proposed method outperforms baselines in multiple evaluation attributes.
- Up till now, REMEDI has been used widely as a representation edit (and probing) approach, and is included in many of the popular model/representation/etc edit benchmarks. I’m actually surprised to see REMEDI when reviewing for COLM today — I have the impression that this paper was in last year’s ICML.

**Reasons To Reject:**

In the context of 2024 April, representation-based model interpretation and control is not as novel as a year ago. However, the novelty in the context of 2023 April is good, which is the time I first saw REMEDI. At that time, there was no RepE (on arXiv in 2023 October) or ReFT (on arxiv in 2024 April).

---

> ### Author Rebuttal · Authors · 2024-05-31
>
> Thanks for the positive review!
>
> Regarding RepE and ReFT (and other representation-level interpretability work), you are right that these are related and we’ll make sure to include a discussion of these in the camera ready version. That said, we'd like to clarify that both RepE and ReFT are substantially different from REMEDI in several ways: RepE (which actually evaluates several probe training methods, both new and existing) is focused on high-level phenomena like "truthfulness" and can't be straightforwardly applied to low-level factual content like REMEDI can. ReFT is a fine-tuning method and requires supervised training data for the update of interest, while REMEDI only requires the description of a new fact. (Please also note that ReFT was published after the COLM submission deadline.)
>
> Questions:
> 1. **Why does the norm of GPT-J’s reps grow substantially at layer 12?** To be fully transparent, we are not sure! We looked at this for some time, but couldn’t reach a satisfying conclusion. As it's not a focus of our paper, we think it’s an interesting topic for future work, as it seems to be true of other language models too.

---

> > ### Comment · Reviewer_idDy · 2024-06-04
> >
> > Thanks for the response. Those indeed make the contributions clearer.

---

### Official Review · Reviewer_oaC1 · 2024-05-03

**Rating:** 8
**Confidence:** 3
**Ethics Flag:** 1

**Summary:**

The paper introduces a technique to edit an LM's representation of an entity, in a given generation context, making it assign high probability to response/generations that are coherent with certain known facts about that entity (these facts are themselves expressed through natural language).

The technique is based on a local edit to a hidden state (local to a layer and position in the sequence) using a trainable affine transformation (of the entity and attribute representations) whose parameters are adjusted using supervision of the kind (generation context, entity position, verbalised attribute/fact, coherent response). The optimisation objective is a combination of losses, which generally aim at encoding (or emphasising) the selected attribute in the entity's representation (e.g., measured by increased probability of a coherent response).

The technique (REMEDI) was shown effective against reasonable baselines in various settings (such as controlled generation, updating factual knowledge, and error detection). Experiments were performed using GPT-J (main body), GPT2-XL and Llama-2-13b (appendix).

While I have a couple of clarification questions and can imagine one or two experiments that would make interesting additions to the paper, I find the current version clear and reasonably complete.

**Questions To Authors:**

1. It's possible that different entities are better edited at different layers, isn't it? Yet, I think for this method we need to choose a specific one, in order to train the probe. Is this a reasonable requirement? For example, wouldn't this potentially differ across entities, entity types, attribute, and many other factors (regardless of my ability to name and enumerate them)? Could you please comment on this?

2. It's not clear to me how you go about multiple token entities.

3. Now, thinking about both these points more generally: what's the reason for working with local edits? When I say 'local' I do mean specific to a layer and position (I get why it's local to a generation context, as opposed to global to the entire architecture). Is it a technical motivation, perhaps something about the probe, or something else? Couldn't, for example, the edit be say a trainable prefix that affects the entire Transformer stack?

4. I get that the controllable generation setting is only one setting (next to the other two) used to demonstrate the approach. But, still, I'd like to hear more about it in relation to a method for controllable generation of the kind [Plug&Play LM](https://openreview.net/pdf?id=H1edEyBKDS). I'm not per se asking for an experiment (though adding it to your table is probably not too difficult for a final revision), but I'm curious to understand how the techniques relate and their relative pros and cons.

**Reasons To Accept:**

Clear and complete paper, addressing a relevant topic, which is likely interesting to COLM's audience (and beyond). The paper discusses limitations in an open and insightful way (perhaps with a couple of points that could be emphasised more and made clearer, and which I document later).

A simple technique, shown effective across a few interesting applications for the ability to edit/control knowledge locally to a given generation context.

**Reasons To Reject:**

I can list some things that can be made clearer, but I don't think these are reasons to reject the paper. Some of these are small clarification points (ie, addressable in camera-ready) others may inspire a future revision of this work or future papers.

1. The edits being local to a layer seem to force the designer to choose a layer, for any edit that will ever be made, using some development data/criterion, and it's not immediately obvious (to me at least) that this choice should (in principle) be made independently of the entity and attribute we intend to edit/control. I think this could have been discussed more.

2. The edits being local to a position seem to force the designer to choose a token to 'stand for the entity'. It's unclear to me whether this means we are limited to working with single-token entities (i.e., one or more word tokens which are split into multiple subword units, or whatever the LM token definition is).

3. The response and the factual statement (~ verbalised attribute) are single strings (ie, one string each). It's my impression that this is not a limitation of the method itself (the optimisation objective could be easily expressed for multiple linguistic realisations of the attribute and multiple coherent responses) but just a reflex of the data available to the authors. It would be interesting to use paraphrases (even if automated) of the factual statement and/or response to assess the method's ability to handle and/or reproduce this kind of variability (and perhaps also to inform the edits themselves).

---

> ### Author Rebuttal · Authors · 2024-05-31
>
> Thank you for the detailed review and constructive feedback! We agree with many of your clarifications and will touch up our discussion in the final version to address all of these points.
>
> To briefly answer your questions:
> 1. **Which layer?** It is very possible that different entities prefer to be edited at different layers of the model. When averaging across an entire task (factual editing vs. incoherence), we found modest differences across layers for factual editing and little difference at all for incoherence (Figure 3 in appendix). It’s also worth noting that the results of Hase et al. 2023 suggest you should be able to successfully perform factual editing anywhere in the network.
> 1. **Multiple token entities:** We always edit the last token of multi-token entities, in line with what other factual editing methods do (e.g., ROME/MEMIT). In preliminary experiments, we didn’t find substantial differences when editing other tokens, but the last token seemed to be most effective.
> 1. **Why local edits?** It’s a very good question! Ultimately, local representation editing is designed around a theory of how LMs encode state in their representations. This framework serves both **interpretability goals** (how do LMs resolve information described in the context with information from pretraining?) in addition to practical goals like controllable generation. We illustrate one advantage of locality by repurposing REMEDI as a probe, but we also think local methods have promise for further analysis techniques that help triangulate errors encoded in LM representations.
> 1. **Comparison to controllable generation:** We agree this is important for us to discuss. We’ll look into additional baselines, but at a minimum we’ll add some discussion to our related work.
>
> References:
> Hase et al, 2023. Does Localization Inform Editing? Surprising Differences in Causality-Based Localization vs. Knowledge Editing in Language Models

---

### Official Review · Reviewer_zhP2 · 2024-05-12

**Rating:** 7
**Confidence:** 5
**Ethics Flag:** 1

**Summary:**

This work proposes a new knowledge editing approach called REMEDI, which enriches entity representations with given facts (attributes). Specifically, REMEDI creates a new entity representation (edit vector) by combining an attribute using an affine transformation. Importantly, this approach does not require updating LM parameters, unlike other knowledge editing methods such as vanilla finetuning, ROME, and MEND. Also, it would be more straightforward to apply in different experimental settings compared to ROME, which requires causal tracing, and MEND, which requires tuning a hypernetwork.

The experiments demonstrate that REMEDI is effective in controlling output sequences. First, it successfully decouples strong priors (e.g., occupation) associated with person names (Section 4.1) and generates correct target sequences. Second, REMEDI shows reasonable performance in knowledge editing, achieving nearly perfect efficacy and a perfect neighborhood score. Additionally, REMEDI can be used to detect failure cases of LMs (Section 5).

**Questions To Authors:**

- Have you considered cases that involve multiple entities? I believe that REMEDI can be applied to those cases easily, and it would be an interesting future direction.
- I think this approach could be efficient and scalable as we can precompute attribute representations. Have you run a performance comparison with other knowledge editing methods?
- [Metadata Shaping paper](https://arxiv.org/pdf/2110.08430) might be related. Perhaps, REMEDI can be seen as a soft version of this.

**Reasons To Accept:**

- This approach addresses shortcomings of prompting approaches (e.g., prepending facts), which are known to be effective, but it might be limited by the max input length of LMs. This work proposes a clever solution for this issue: encoding facts into entity representations without changing the input length.
- This paper is well-written; the technical details and experimental settings/results are clearly explained, and the scope of this work is well managed in the main paper.

**Reasons To Reject:**

- By its design, REMEDI doesn’t update LM parameters. This won’t be a problem in the short term, but it could become an issue when LMs’ knowledge is outdated (e.g., people will change occupations/positions). In such cases, repeatedly applying REMEDI might be cumbersome.
- Note that this is a common issue of knowledge editing benchmarks, but the experimental setup is too clean (e.g., based on KG triples). It would be nice to see how this approach behaves in a bit more wild setting.

---

> ### Author Rebuttal · Authors · 2024-05-31
>
> Thank you for the positive review!
>
> We agree that knowledge editing settings (with fact triples) are a very controlled setting. To clarify, we do additionally include results for Bias in Bios, which comprises professional biographies scraped from the internet. The attributes in this dataset are much more “in the wild” and provide some evidence that REMEDI can handle those attributes.
>
> To answer your questions:
> 1. **Multiple entities?** We didn’t look at multi-entity setups while working on REMEDI, but we agree this is an interesting direction for future work.
> 1. **Performance of REMEDI vs. knowledge editors?** We didn’t explicitly measure runtime/efficiency, but REMEDI should be substantially faster than ROME because once the editing model is trained only a forward pass of the LM is required. We’ll include a short discussion of these differences in the camera ready version.
> 1. **Metadata shaping paper:** Thanks for the pointer! We’ll take a look and include a discussion in the final version where relevant.

---

### Decision · Program_Chairs · 2024-07-10

**Decision:**

Accept

**Comment:**

There is a consensus among the reviewers that the works introduces a novel, interesting and conceptually simple method to editing knowledge in LM. There are certain limitations of the approach, e.g., the application requires inference time modification of the computation, detection of the relevant entity at inferences, the somewhat restricted set-up (assuming fact triples), but these did not prevent the reviewers from recommending acceptance.  Furthermore, these limitations are not unique to the proposed method, which also offers advantages like the relatively simple training procedure.